# Effect of Ultrasonic Treatment on the Physicochemical Properties of Bovine Plasma Protein-Carboxymethyl Cellulose Composite Gel

**DOI:** 10.3390/foods13050732

**Published:** 2024-02-28

**Authors:** Liyuan Wang, Yu Ma, Ruheng Shen, Li Zhang, Long He, Yuling Qu, Xiaotong Ma, Guoyuan Ma, Zhaobin Guo, Cheng Chen, Hongbo Li, Xiangying Kong

**Affiliations:** 1College of Food Science and Engineering, Gansu Agriculture University, Lanzhou 730070, China; wly04122022@163.com (L.W.); mayu_yyds@163.com (Y.M.); ruhengshen@163.com (R.S.); 18215284414@163.com (L.H.); quyuling0106@163.com (Y.Q.); maxiaotong17@163.com (X.M.); maguoyuangsau@163.com (G.M.); guozhb007@163.com (Z.G.); chenchengmlj@163.com (C.C.); 2Institute of Animal Husbandry, Xinjiang Academy of Animal Husbandry, Xinjiang 830011, China; lihb8228@126.com; 3Haibei State Agricultural and Animal Husbandry Comprehensive Service Center, Haibei 810299, China; 13897309088@163.com

**Keywords:** ultrasound, bovine plasma protein, carboxymethyl cellulose, composite gel, gel properties

## Abstract

In order to improve the stability of bovine plasma protein-carboxymethyl cellulose composite gels and to expand the utilization of animal by-product resources, this study investigated the impact of different ultrasound powers (300, 400, 500, 600, and 700 W) and ultrasound times (0, 10, 20, 30, and 40 min) on the functional properties, secondary structure and intermolecular forces of bovine plasma protein-carboxymethyl cellulose composite gel. The results showed that moderate ultrasonication resulted in the enhancement of gel strength, water holding capacity and thermal stability of the composite gels, the disruption of hydrogen bonding and hydrophobic interactions between gel molecules, the alteration and unfolding of the internal structure of the gels, and the stabilization of the dispersion state by electrostatic repulsive forces between the protein particles. The content of α-helices, β-turns, and β-sheets increased and the content of random curls decreased after sonication (*p* < 0.05). In summary, appropriate ultrasound power and time can significantly improve the functional and structural properties of composite gels. It was found that controlling the thermal aggregation behavior of composite gels by adjusting the ultrasonic power and time is an effective strategy to enable the optimization of composite gel texture and water retention properties.

## 1. Introduction

Blood is the primary product in animal slaughter and bovine blood is a valuable protein resource with a protein content ranging from 17% to 20%. Among its components, plasma constitutes 65% of the total blood, containing approximately 8% protein [1]. Bovine blood plasma protein powder is obtained by anticoagulation treatment, separation from plasma, and subsequent processing or drying. It possesses an inherent odor and appears as a pale or light pink powder [2]. The protein demonstrates excellent emulsification, foaming, and thermal gelation properties, making it suitable as a fat substitute in foods, especially in low-fat meat products and emulsified sausages [3]. The gelation properties of bovine blood plasma protein also contribute to enhanced water retention, texture characteristics, and nutritional value in heated products [4]. Additionally, the gel formed by bovine blood plasma protein can act as a carrier for transporting proteins, enhancing water solubility and facilitating transportation [5]. However, the compact structure and limited functionality of singular bovine blood plasma protein systems restrict their applications in the food industry. Hence, the combination of bovine blood plasma protein with polysaccharides is essential to improve its surface activity and gelation properties [6].

Carboxymethyl cellulose (CMC) is a water-soluble anionic cellulose derivative, primarily existing in sodium salt form. Since the 1940s, CMC has been widely applied in the food industry due to its outstanding thickening, suspending, and stabilizing properties [7]. Introduced to cellulose chains by -CH_2_COOH groups, CMC typically exhibits a general average degree of substitution (DS) ranging from 0.5 to 1.5 [8]. CMC serves as a hydrophilic colloidal material, allowing the preparation of transparent water gels with excellent mechanical and adhesive properties. In the food industry, CMC hydrogels are commonly used to improve the texture and mouthfeel of food and beverages. They are also employed as edible gels in products such as bread and confectionery [9]. Therefore, CMC is a representative polysaccharide with abundant content and extensive applications in the food industry. Polysaccharides and food proteins, due to their large polymeric structures, are often used as natural gelling agents, providing systematic opportunities for the development of gel-like biomaterials with adjustable physicochemical and functional properties [10]. The interaction between proteins and polysaccharides offers a platform for the development of diverse functionalities through modification techniques, not only in food, but also in the health industry. By controlling the interactions between proteins and polysaccharides and employing modification techniques, these biomaterials can be diversified in terms of functionality [11]. Numerous studies have investigated the interaction between proteins and polysaccharides and their physical modification techniques in solution or gel systems (thermal or cold), providing a wide range of potential combinations and broadening the application scope of proteins and polysaccharides in the food industry.

A number of physical methods have been used to modify the structure of proteins and polysaccharides and enhance the functional properties of protein-polysaccharide gels. Ultrasound (20–100 kHz) is an efficient, economical, safe, and operationally simple physical method that can be used to modify the gel properties of proteins and polysaccharides in the food industry [12]. When ultrasound passes through a fluid, cavitation occurs, resulting in physical effects (high pressure, shear, microstreaming shock waves), chemical effects (free radicals), and thermal effects [13]. These effects induce changes in protein structure, further altering the gel properties of certain food proteins. Ultrasound has found wide applications as an auxiliary technique in industries such as food and biotechnology [14]. Studies by Hu et al. [15] discovered that ultrasound treatment at 20 kHz and 400 W promotes the formation of a three-dimensional gel structure, making the calcium-ion-induced soy protein isolate (SPI) structure denser and more uniform. This enhances gel water-holding capacity and gel strength. Zhao et al. [16] found that ultrasound treatment reduces particle size, and exposes hydrophobic and free thiol groups, aiding in the formation of dense and uniform gel networks in potato protein isolates and potato protein/egg white mixtures. Shen et al. [17] observed that ultrasound treatment significantly improves the storage modulus (G′), hardness, and water-holding capacity (WHC) of glucose-δ-lactone-induced whey protein gels. Additionally, it increases β-folding and decreases α-helix, thereby enhancing the gel strength of fish mince. Similarly, Gao et al. [18] noted that ultrasound treatment forms more non-covalent bonds and disulfide bonds, resulting in a dense gel structure and significantly improving the WHC of fish mince gels.

However, no studies have reported the effects of ultrasound treatment on the properties of composite gels made from bovine plasma protein and carboxymethyl cellulose. Therefore, the purpose of this study is to investigate the impact of ultrasound power (300 W, 400 W, 500 W, 600 W, and 700 W) and ultrasound time (0, 10 min, 20 min, 30 min, and 40 min) on the structure and properties of the composite gel. Structural analysis of ultrasound-treated bovine plasma protein-carboxymethyl cellulose composite gels was conducted using surface hydrophobicity, free thiol groups, zeta potential, Raman spectroscopy, Fourier-transform infrared spectroscopy (FTIR), and scanning electron microscopy (SEM). The gel properties of the ultrasound-treated composite gels were determined through rheological, thermodynamic, and WHC measurements. In this study, the mechanism of ultrasound on protein-polysaccharide composite gels was investigated to provide a theoretical basis for the development and application of bovine plasma protein-carboxymethylcellulose composite gels in the field of novel foods.

## 2. Materials and Methods

### 2.1. Materials

Fresh bovine blood was purchased from a local slaughterhouse in Gansu, China. Fresh bovine blood was obtained via centrifugation at 10,000 rmp, 4 °C, for 10 min using a TG16-WS high-speed centrifuge (Shanghai Lu Xiangyi Centrifuge Instrument Co., Ltd., Shanghai, China). The separated bovine plasma was freeze-dried using a LGJ-10 freeze-dryer (Songyuan Huaxing Technology Co., Ltd., Beijing, China) (freezing temperature: −80 °C, 24 h; vacuum: 0.015 Bar), and named as bovine plasma protein powder (BPP) (protein content: 81.27%; fat content: 1.95%; moisture content: 7.83%; ash content: 8.32%), and then kept in a cooled place to keep in reserve. Carboxymethyl cellulose was purchased from Henan Wanbang Chemical Technology Co., Ltd. in China; potassium bromide (KBr) was spectroscopically pure, and all other chemical reagents were analytically pure and purchased from Shanghai McLean Biochemical Co., Ltd., Shanghai, China. Tris-Gly buffer (pH = 8, 100 mmol/L Gly, 4 mmol/L EDTA, 100 mmol/L Tris), Ellman’s reagent (4 mg/mL of 5′,5-dithiobis-(2-nitrobenzoic acid) dissolved in Tris-Gly buffer).

### 2.2. Sample Preparation

A total of 7% bovine plasma proteins and 0.6% carboxymethyl cellulose were added to a certain amount of distilled water and stirred well, and homogenized in an ice-water bath for 2 min using a FJ-200 high-speed dispersion homogenizer (Shanghai Specimen Model Factory) at a speed of 12 kr/min. The mixture was then stirred in a HJ-1 magnetic stirrer (Changzhou Surui Instrument Co., Ltd., Changzhou, China) for 10 min, and then after that, the samples were processed using a JY92-IIN ultrasonic treatment processor (Ningbo Scientz Biotechnology Co., Ltd., Ningbo, China) with a 6 mm high-grade titanium tip; the samples were in an ice bath (the sample ultrasound temperature was controlled below 25 °C). The ultrasonic power was 300 W, 400 W, 500 W, 600 W and 700 W, and the durations were 0, 10, 20, 30 and 40 min in pulse mode (3 s on and 3 s off). The samples of each group were added into 10 mm tubes and heat treated at 85 °C for 40 min, rapidly cooled in an ice bath and left at 4 °C for 16 h. The control test without ultrasound treatment (ultrasound time 0 min) was used as a blank sample.

### 2.3. Gel Strength Test

Gel strength was assessed according to the method described by Wang et al. [19]. Gel strength was assessed using an instrumental weave analyzer (TA.XT Express, Stable Micro Systems) equipped with a P/0.5R test probe. The test parameters were set as follows: trigger force of 5 g; prediction speed of 1.0 mm/s; test speed of 1.0 mm/s; post-test speed of 1.0 mm/s; and puncture distance of 8.0 mm. The gel strength analysis was performed using the Exponent Connect 8.0.7.0 software that came with the instrument.

### 2.4. Determination of Water Holding Capacity (WHC)

Water retention measurements of samples after various treatments were performed according to the method outlined by Fang et al. [20] with minor modifications. Composite gel samples (3 g per tube) were centrifuged in 5 mL centrifuge tubes at 8000 R/min for 10 min at 4 °C. After centrifugation, the tubes were inverted and drained, and the water droplets remaining on the inner wall were removed with dry filter paper. The weight of the tube containing the gel sample before and after centrifugation was accurately measured and the water holding capacity (WHC) of the composite gel was calculated according to the following equation:WHC = Wr/Wt × 100(1)
where Wt—total weight of gel sample tubes loaded before centrifugation, Wr—weight of gel sample tubes loaded after centrifugation.

### 2.5. Rheological Behaviors

Following the procedure outlined by Chen et al. [21], a DHR-1 rheometer (TA Instruments, New Castle, DE, USA) with a 40 mm parallel geometric plate was utilized. A volume of 2 mL of composite gel solution was introduced into the gap between the parallel plates, with silicone oil applied to the sample’s perimeter to mitigate water evaporation. The heating rate was set at 2 °C/min, the cooling rate at 5 °C/min, and the oscillation frequency at 0.1 Hz, while maintaining a stress of 0.1%. The heating/cooling temperature slope experiment was conducted within the range of 25 to 85 °C.

### 2.6. Thermostability (DSC)

Following the procedure outlined by Bi et al. [22], a Differential Scanning Calorimetry (DSC) instrument was employed to investigate the enthalpy of the gels. The gel was accurately weighed and placed into a sealed aluminum dish. An empty crucible served as the control group. The temperature was increased within the range of 20 to 180 °C, with a temperature rise rate of 10 °C/min. The peak temperature (Tp) and enthalpy (ΔH) were determined from the DSC heat flow curves.

### 2.7. Intermolecular Force

#### 2.7.1. Determination of Surface Hydrophobicity

The gels were diluted tenfold with PBS buffer, followed by homogenization and centrifugation. The protein concentration of the supernatant was adjusted to 0.3 mg/mL. Subsequently, 2 mL of the diluted sample was combined with 40 μL of ANS solution (sodium 8-anilino-1-naphthalenesulfonate dissolved in ethanol, 1 mmol/L) and incubated in the dark for 10 min. Finally, surface hydrophobicity analysis was performed using a Hitachi F-7000 Flourescence Spectrofluorometer (Hitachi, Tokyo, Japan) with excitation and emission wavelengths set at 380 nm and 485 nm, respectively [23].

#### 2.7.2. Determination of Free Sulfhydryl

The gel was diluted tenfold with PBS buffer, followed by homogenization and centrifugation. The protein concentration of the supernatant was determined using the BCA method. Subsequently, the supernatant was further diluted fourteenfold with Tris-Gly buffer, and 0.1 times the volume of Ellman’s reagent was added. The mixture was heated at 40 °C for 40 min to complete the reaction. Finally, the sample’s free sulfhydryl content was analyzed using a Thermo K3 enzyme mark instrument (Thermo Fisher Scientific, Shanghai, China) at 412 nm. PBS buffer was used as the reagent blank. The free sulfhydryl content of TUEP and UEP was calculated using the following formula: μmol SH/g protein = (15.1 × A412 × 73.53)/C, where 15.1 is the dilution factor ((1 + 14 + 0.1)/1), A412 is the absorbance, 73.53 is obtained by unit conversion (106/(1.36 × 104) M^−1^·cm^−1^), and C is the protein concentration (mg/mL) [24].

#### 2.7.3. Determination of Zeta Potentials

The zeta potentials of the gel (1 mg/mL) were determined at room temperature (25 °C) using a Zeta Nano Analyzer supplied by Brookhaven Instruments Co., Ltd. (Brookhaven, GA, USA), following the procedure outlined by Guan et al. [25].

#### 2.7.4. Raman Spectra

The Raman spectra of the gels were acquired using a High-Speed and High-Resolution Confocal Microscope Raman Spectrometer supplied by HORIBA Scientific Instruments CO., Ltd. (LabRAM Odyssey, Paris, France), employing an excitation wavelength of 532 nm at room temperature (25 °C) [26].

### 2.8. Fourier Transformed Infrared (FTIR) Analysis

A Fourier transform infrared spectrometer detector (Tensor 27, Bruker, Germany) was utilized to analyze the secondary structure of the gels. A small amount of KBr was added to the freeze-dried samples, mixed, ground into powder, and compressed into a tablet using a tablet press. The parameters were configured as follows: wavenumber range of 4000–400 cm^−1^; resolution of 4 cm^−1^; and 32 scanning times. The background spectrum was recorded prior to measuring the sample [27].

### 2.9. Microstructure Analysis

The microstructures of the gels were measured with a Hitachi SU8010 SEM (Hitachi Ltd., Tokyo, Japan). The gels were immobilized overnight in 2.5% glutaraldehyde, followed by rinsing with PBS buffer. Subsequently, the samples were dehydrated using ethanol (60–100% (*v*/*v*)) and freeze-dried with a vacuum freeze dryer. Finally, the samples were gold-sputtered and observed under a 5 kV accelerating voltage at 5000 magnification [28].

### 2.10. Statistical Analysis

All statistical analyses were performed using SPSS 18 data analysis software (IBM, New York, NY, USA). Data are shown as mean ± standard deviation. One-way analysis of variance (ANOVA) was used to determine the significance of the main effect and Duncan’s multiple range test was used to determine significant differences between groups (*p* < 0.05). Two-way ANOVA was used to analyze the interaction between factors. *p* < 0.05 was considered a significant difference in the models. Graphs were generated using Origin 8.5 software (Origin Lab Corp., Northampton, MA, USA). All tests were repeated three times.

## 3. Results and Discussion

### 3.1. Gel Strength

Gel strength is an important indicator reflecting the gelatinous properties and texture of gel products [29]. The gel strength of bovine plasma protein-carboxymethyl cellulose composite gel is shown in Figure 1a. The gel strength of the composite gels with an ultrasound time of 30 and 40 min were significantly higher than that of the control group (*p* < 0.05) and, except for the treatment group with an ultrasound power of 300 W, the gel strength of the composite gels reached its maximum at the ultrasound time of 30 min in the other groups when the ultrasound power was the same. The increase in the strength of the composite gel may be related to the action of ultrasound, which generates microfluidization and cavitation, and promotes covalent cross-linking between the protein molecules in the composite gel under the action of thermal processing [30]. The increased degree of cross-linking between protein molecules during heating enhances the stability of the composite gel, promoting the gel strength of the bovine plasma protein-carboxymethyl cellulose gel. However, excessive ultrasound power can lead to the formation of insoluble aggregates disrupting the dense and ordered network structure of the composite gel, thereby reducing the strength of the composite gel [31]. Zou et al. [32] found that the plasma gel strength of chickens without ultrasonic treatment was 1.32 N, which was significantly lower than sonicated gel alone and combined sonicated gel with konjac glucomannan addition (*p* < 0.05); the plasma gel strength of chickens after ultrasonic treatment was 108.3% higher than that of the control group.

### 3.2. Water Holding Capacity (WHC)

Changes in the water-holding capacity (WHC) of protein gels can characterize their water retention, which is an important indicator of composite gel [33]. The WHC is related to the gel structure of the protein and the state of water–protein interaction. The water-holding capacity (WHC) of the bovine plasma protein-carboxymethyl cellulose composite gel samples is shown in Figure 1b. The WHC of the control group was the lowest, and the water retention of the composite gel significantly increased after ultrasound treatment (*p* < 0.05). When the ultrasound time was 30 min and the ultrasound power was 500 W, the water-holding capacity of the composite gel was the strongest, increasing by 27.39% compared to the control group. Ultrasound has a positive effect on the water retention of the bovine plasma protein-carboxymethyl cellulose composite gel because the microstreaming and cavitation effects of ultrasound promote the expansion and conformational changes in plasma proteins, leading to the extension and folding of protein structures. This enhances the gel network’s ability to trap and retain water, forming a denser and more stable gel structure [34]. Additionally, the exposure of hydrophobic residues caused by ultrasound treatment may facilitate the formation of protein–protein aggregates during the gelation process, thus forming a gel network with better WHC. However, excessive ultrasound power can result in the formation of some insoluble aggregates, making the gel network more porous, which is not conducive to the binding of water molecules to the gel network, resulting in an initial increase followed by a decrease. Zhang et al. [35] made a similar observation, noting that the WHC of myofibrillar protein gels gradually increased from 37.95% to 49.09% as the ultrasound power was increased from 0 W to 400 W, and that the WHC significantly decreased to 39.02% when the ultrasound power was increased from 400 to 1000 W (*p* < 0.05).

### 3.3. Rheological Behaviors

Rheological properties are closely related to functional properties of proteins, such as the ability to gel. The storage modulus (G′, Pa), loss modulus (G″, Pa), and phase angle (tanδ) are commonly used to represent the viscoelastic and structural characteristics of proteins. Generally speaking, the energy stored during gel formation due to elastic deformation is G′, which can be used to indicate its solid properties. The energy lost due to viscous deformation is G″, which can be used to describe its liquid properties. Figure 2 illustrates the thermal gelation temperature gradient curves of the composite gel system under different ultrasound powers and durations. In the temperature range of 25–40 °C, both the storage modulus (G′) and loss modulus (G″) remain constant or slightly decrease. This might be because this temperature is insufficient to fully extend the protein structure, the reactive groups are not exposed in large quantities, and there is limited intermolecular cross-linking between protein molecules [36]. As the temperature exceeds 40 °C, protein aggregates gradually form, and the gel matrix starts to strengthen, leading to a rapid increase in G′ and G″. During the cooling stage (85–25 °C), the G′ values of all samples sharply increase. This is attributed to cooling promoting the strengthening or formation of both covalent and non-covalent bonds [37]. Cooling enhances the hydrogen bond formation and stabilizes the gel structure, significantly increasing G′, as reported [38]. The terminal value of G′ at the end of the cooling stage typically represents the elasticity of the finally formed gel [20]. The loss angle tanδ reflects the dynamic changes in proteins during heating. In the early heating stage, tanδ for all treatment groups is greater than 1, while in the later heating stage, tanδ is less than 1, indicating that gelation gradually completes during the later heating stage forming gels with typical elastic characteristics, where G′ > G″. The G′ of ultrasound-treated composite gels is higher than that of untreated ones, indicating that ultrasound promotes the formation of the gel network and the development of good gel elasticity. This may be due to ultrasound exposing more hydrophobic groups and free thiol groups, enhancing hydrophobic interactions and disulfide bonds, thus improving the elasticity of the composite gel [32]. Previous studies by Zhao et al. [16] reported that ultrasound treatment increased the elasticity of gelatinized insoluble potato protein isolates, and Li et al. [39] found that ultrasound treatment enhanced the elasticity of chicken myofibrillar protein gels, possibly due to the exposure of hydrophobic groups and free thiol groups contributing to the formation of a more stable gel network structure.

### 3.4. Thermostability (DSC)

In DSC results, the peak temperature (Tp) is the denaturation temperature of composite gel, which represents the thermal stability of protein gel. The enthalpy change (ΔH) is the ordered structure of a protein and represents the degree of aggregation of protein molecules [40]. As can be seen from Table 1, both Tp and ΔH of the sonicated composite gel were elevated compared to the unsonicated composite gel, and the ΔH values of the samples with sonication times of 20 and 30 min and sonication powers of 400, 500, and 600 W were significantly higher than those of the unsonicated samples (*p* < 0.05). At the same power, the ΔH value of the composite gel will reach its maximum at 30 min of sonication time; when the ultrasonic power is 500 W and the ultrasonic time is 30 min, the ΔH is maximum 203.82 J/g, and when the sonication time is extended from 30 to 40 min, the ΔH value of the composite gel decreases. It indicates that proper sonication can improve the thermal stability of the composite gel, while when the ultrasound time is too large it can destabilize the protein gel. Overall, sonication helps to maintain protein stability and structural integrity. The higher ΔH values indicate the development of intermolecular interactions during the gelation process of the composite gel, forming a more stable gel network structure. As a result, more energy and higher temperatures are required to break up highly cross-linked structures [41]. Similarly, Mozafarpour et al. [42] found that the ΔH of the grass bean protein emulsion gel prepared by ultrasound increased, and the heat resistance of the gel network was enhanced, and confirmed the development of the intermolecular interaction during the formation of emulsion gel by ultrasonic treatment.

### 3.5. Intermolecular Force

#### 3.5.1. Surface Hydrophobicity

The hydrophobicity of a surface is related to the conformation, size, amino acid composition and sequence of the protein, as well as the intra- and inter-molecular connections, and can be used to study changes in the tertiary structure of proteins [43]. As shown in Figure 3a, with the prolongation of ultrasound time, the surface hydrophobicity first showed an increasing trend and the surface hydrophobicity of the composite gel reached the maximum at 30 min of ultrasound time. After 30 min, the surface hydrophobicity of the composite gel showed a decreasing trend. The composite gel’s minimum surface hydrophobicity at 0 min of ultrasound time is 86.34, while the maximum surface hydrophobicity of the composite gel at 30 min of ultrasound time and 500 W ultrasound power is 137.31. This indicates that ultrasonic treatment increases the surface hydrophobicity of the composite gel. This may be because the cavitation effect generated by ultrasound breaks non-covalent bonds, promotes partial unfolding of the protein, and exposes more hydrophobic groups on the protein molecule surface [44]. When the ultrasound time is too long and the ultrasound power is too strong, the surface hydrophobicity of the protein decreases. This may be due to the excessive generation of heat during the ultrasonic treatment process. During cooling, the hydrophobic interactions of the protein will cause protein aggregation, thus reducing the exposure of hydrophobic groups. These findings are consistent with the results of the research conducted by Cui et al. [45]. These results indicate that appropriate cavitation effects produced by ultrasound treatment can alter the spatial structure of proteins, leading to the disruption of protein hydrogen bonds and intermolecular forces, exposing hydrophobic groups inside the protein, and enhancing surface hydrophobicity.

#### 3.5.2. Free Sulfhydryl

Free sulfhydryl plays an essential role in the structural stability of proteins. By determining the sulfhydryl content, researchers can study changes in protein structure. Under the action of peroxides, sulfhydryl easily form disulfide bonds, thereby improving the gel network structure of proteins [46]. As shown in Figure 3b, the free sulfhydryl content of the composite gels showed a gradual increase as the ultrasound time increased from 0 to 30 min. And the free sulfhydryl content of the composite gel when the ultrasound time reached 20, 30 and 40 min was significantly higher than that of the composite gel without ultrasonic treatment (*p* < 0.05). However, the free sulfhydryl content of the composite gels with an ultrasound time of 40 min was significantly lower (*p* < 0.05) than that of the samples with an ultrasound time of 30 min when the ultrasound power was 400, 600 and 700 W. These data indicate that ultrasonic treatment can increase the content of free sulfhydryl in the composite gel. This is likely due to the shear force generated by cavitation, which unfolds the protein and breaks the disulfide bonds, exposing the sulfhydryl groups within the protein [47]. However, the cavitation effect generates many free radicals, and excessively high ultrasound power may oxidize the exposed free sulfhydryl, leading to a decrease in the content of free sulfhydryl in the composite gel by re-forming disulfide bonds. Additionally, excessively high ultrasound power may lead to the formation of composite aggregates, burying some of the free sulfhydryl [48]. Similarly, Zhao et al. [16] reported that the content of free sulfhydryl in potato protein isolates first increased from about 2.25 µmol/g (0 W) to 6.25 µmol/g (400 W), and then decreased to 3.0 µmol/g (600 W).

#### 3.5.3. Zeta Potential

The Zeta potential can reflect the strength of electrostatic repulsion or attraction between particles in a gelatinous protein system, and is related to gel stability [49]. It is generally believed that the further the Zeta potential is from zero, the more stable the gel system formed, while closer to zero indicates a higher likelihood of flocculation or breakdown [50]. The impact of ultrasonic treatment time on the Zeta potential of bovine plasma protein-carboxymethyl cellulose composite gel is shown in Figure 3c. Both before and after ultrasonic treatment, the Zeta potential of the composite gel is negative, with absolute values ranging from 15 to 25 mV, indicating the presence of a large number of negatively charged amino acids on the protein surface. Furthermore, the samples with a sonication time of 30 min all had absolute values of zeta potential greater than 20 mV. Feng et al. [51] suggest that when the absolute value of the system’s Zeta potential is greater than 20 mV, the colloidal system is stable. Therefore, the samples with a sonication time of 30 min protein gels mentioned above all form relatively stable gel systems. At an ultrasound time of 30 min and an ultrasound power of 500 W, the absolute value of the Zeta potential of the composite gel is the highest (23.53 mV), indicating that at this point, the electrostatic repulsion between protein particles is at its maximum, making aggregation less likely. With increasing ultrasound time, the absolute value of the Zeta potential of the composite gel first increases and then decreases. This phenomenon may be due to the exposure of polar groups in the protein caused by ultrasonic treatment. Similarly, Wang et al. [52] found that the absolute value of ζ potential of soybean protein emulsion gel after high-intensity ultrasound treatment increased significantly, reaching the maximum value at 450 W. He speculated that the increase in the absolute value of ζ potential would enhance the electrostatic repulsion between proteins and inhibit aggregation, thus promoting the stability of the emulsion.

#### 3.5.4. Raman Spectra

The dual Raman spectra around 830 cm^−1^ and 850 cm^−1^ are utilized to monitor the microenvironment surrounding the tyrosine residues. The tyrosine dual peak ratio (I850/I830) serves to detect whether tyrosine residues are exposed to the solvent, with the relative intensity reflecting the nature of hydrogen bonds and the status of phenolic hydroxyl groups in the tyrosine side chain. The number of grams N of tyrosine residues buried inside the molecule and exposed on the surface of the molecule can be calculated according to the equation (0.5Nburied + 1.25Nexposed = I850/I830, Nburied + Nexposed = 1) [53]. Typically, when the tyrosine dual peak ratio (I850/I830) is between 0.7 and 1.0, it indicates that tyrosine residues are buried within the protein molecule. When the ratio is between 0.90 and 1.45, it suggests that tyrosine residues are exposed to the solvent [54]. As shown in Table 2, the tyrosine dual peak ratio (I850/I830) for all composite gels is around 1.0, indicating hydrogen bonding between the OH on tyrosine residues and other neutral groups on the protein. The I850/I830 ratio for the composite gels ranges from 0.9271 to 1.0127; with the increase in ultrasonic power and ultrasonic time, it first increased and then decreased, and reached the maximum value at the ultrasonic power of 500 W and ultrasonic time of 30 min. In addition, N-exposed showed a tendency to increase and then decrease with increasing ultrasonic power and time, and the highest N-exposed (0.6836) was obtained at 500 W, 30 min. This indicates that under appropriate ultrasound treatment, tyrosine residues in protein molecules are predominantly exposed, forming more hydrogen bonds with water molecules. However, at higher ultrasound power and time, I850/I830 significantly decreases possibly due to the high-pressure effect causing the exposed tyrosine residues’ phenolic hydroxyl and hydrophobic groups to embed into the protein molecules during the protein aggregation process, resulting in reduced Raman intensity [26]. Wang et al. [55] found that I850/I830 of soybean protein separate-tannic acid complex increased first and then decreased with the increase in ultrasonic power, and the peak value of I850/I830 was 1.102 at 450 W. It is shown that moderate ultrasonic power contributes to the exposure of tyrosine residues as hydrogen bond donors or acceptors to polar microenvironments.

### 3.6. Fourier Transformed Infrared (FTIR) Analysis

FTIR is a commonly used method for studying the secondary structure and functional groups of proteins. Figure 4. illustrates the FTIR spectra of composite gels treated with different ultrasonic powers. The vibrational states of chemical bonds in proteins correlate with FTIR. The shape and position of peaks in the infrared spectrum show no significant changes, indicating that ultrasonic treatment does not completely break or form protein functional groups. Proteins in the composite gel exhibit multiple characteristic absorption peaks in the infrared region, including the amide I band (1700~1600 cm^−1^) mainly generated by C=O stretching vibration, the amide II band (1600~1500 cm^−1^) caused by N-H bending and C-N stretching vibrations, the amide A band (near 3300 cm^−1^) arising from N-H stretching vibration, and the amide B band (3000~2900 cm^−1^) induced by C-H stretching vibration and NH_3_ [56]. The intensity of the amide A peak in the composite gel subjected to ultrasonic treatment is significantly higher than that in the control group, suggesting that ultrasonic treatment can slightly enhance the hydrogen bonds in the composite gel. The intensity of the amide B peak in the composite gel increases slightly under ultrasonic treatment, indicating an increase in the content of methyl and methylene groups. Similarly, Xue et al. [46] found that the intensities of the A and B peaks in the gel of egg white protein significantly increased under ultrasonic treatment.

The amide I band associated with the stretching vibration of C=O is used to characterize the protein’s secondary structure, including β-sheet (1600–1640 cm^−1^), random coiling (1640–1650 cm^−1^), α-helix (1650–1660 cm^−1^), and β-turn (1660–1700 cm^−1^) [52]. As shown in Table 3, the quantities of β-sheet, α-helix, and β-turn in the composite gel subjected to ultrasonic treatment are significantly higher than those in the control group, while the quantity of random coiling is significantly lower. α-helix and β-turn are generally highly ordered and relatively stable secondary structures, and β-sheet is crucial for maintaining the main structure of the gel [57]. Changes in the α-helix content may reflect alterations in hydrogen bonds between carbonyl (C=O) and amino (N-H) groups in proteins, suggesting that the increase is possibly due to the maintenance of α-helix by hydrogen bonds formed between the carbonyl and amino groups of the peptide chain [35]. The increase in β-sheet may be attributed to ultrasonic treatment promoting cross-linking between polysaccharides and proteins. This indicates that ultrasonic treatment transforms the structure of the composite gel towards higher orderliness, consistent with macroscopically forming a three-dimensional network structure in the gel, leading to increased gel strength. Interestingly, the results for α-helix and random coiling in this study are opposite to those observed in the gel of porcine myofibrillar protein under ultrasonic treatment [44], suggesting that the effects of ultrasonic treatment may vary for different proteins.

### 3.7. Microstructure Analysis

The impact of different ultrasound treatments on the microstructure of bovine plasma protein-carboxymethyl cellulose composite gels is illustrated in Figure 5. From the figure, it is evident that the gel structure of the untreated composite gel is disordered, and the pores are irregular. Ultrasound pretreatment improves the microstructure of the gel to some extent, transforming the gel’s spatial structure from disorder to order, reducing the gel pore size, and thickening the gel pore walls. Under the same ultrasound power, with increasing ultrasound time, the gel structure transitions from disorder to order, and the gel walls become thicker and more substantial. Under the same ultrasound time, increasing ultrasound power shows a similar trend. The mechanical and cavitation effects of ultrasound enhance the cross-linking interaction between protein molecules in the composite gel compared to the control group. This results in a denser and more uniform gel network structure, with stronger water-binding capacity within the network. The reduction in protein particle size and exposure of functional groups on protein surfaces due to cavitation effects under ultrasound facilitate better interactions between protein molecules, leading to increased cross-linking density and the formation of a more refined and orderly gel network structure [58]. This finely organized gel network structure contributes to improved gel properties. However, excessively high ultrasound power may cause protein denaturation, weakening interactions between protein molecules, and the disruption of hydrophobic interactions among proteins, resulting in uneven distribution of pores in the gel network and a decrease in mechanical performance. Similarly, Zou et al. [32] found that after ultrasonic treatment combined with konjac glucomannan, the protein gel surface was smooth, the chicken plasma protein concentration was good, and a fine, uniform and stable gel network structure was formed.

## 4. Conclusions

This study investigated the influence of ultrasonic treatment (300 W, 400 W, 500 W, 600 W, and 700 W) and ultrasonic time (0 min, 10 min, 20 min, 30 min, 40 min) on the structure and gel properties of bovine plasma protein-carboxymethyl cellulose composite gel. The results indicate that ultrasonic treatment promoted the transformation of protein conformation from random curling to α-helix, β-turn, and β-sheet, increased the surface hydrophobicity and free sulfhydryl group content, and enhanced the hydrophobicity and disulfide bond in the composite gel. Changes in protein conformation and increased intermolecular forces result in the formation of a dense and uniform composite gel network, leading to improved rheological properties, gel strength, and WHC of the composite gel. At the same ultrasound power, the improvement in water holding capacity, thermal stability, intermolecular forces, and secondary structure of the composite gel was most significant at 30 min. However, when the sonication time was too long (>30 min), the pores of the reticular structure of the composite gel became larger, and G′, gel strength and WHC decreased. This study demonstrated that ultrasonication is an effective method to improve the properties of bovine plasma-carboxymethylcellulose composite gels, which provides a theoretical basis for their application in food processing.

## Figures and Tables

**Figure 1 foods-13-00732-f001:**
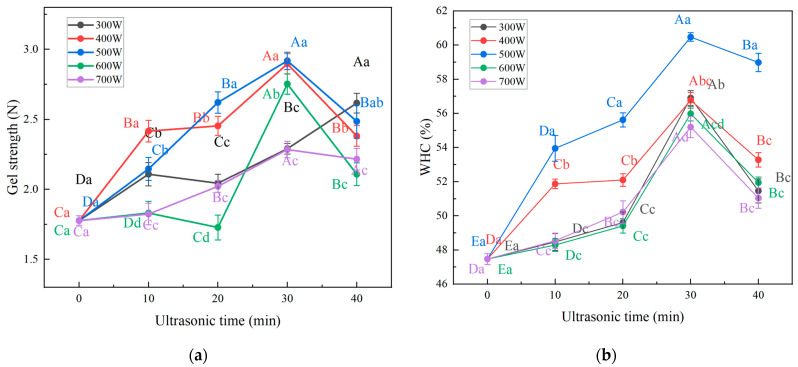
(**a**): Gel strength of bovine plasma protein-carboxymethyl cellulose composite gel under different ultrasonic time and power. (**b**): WHC of bovine plasma protein-carboxymethyl cellulose composite gel under different ultrasonic time and power. Different superscript letters mean significant differences between values with different ultrasound power and ultrasound time (*p* < 0.05). Lower case letters indicate significant differences between different ultrasound powers. Upper case letters indicate significant differences for different ultrasound times. The samples without ultrasonic treatment (ultrasonic time 0 min) were the control samples.

**Figure 2 foods-13-00732-f002:**
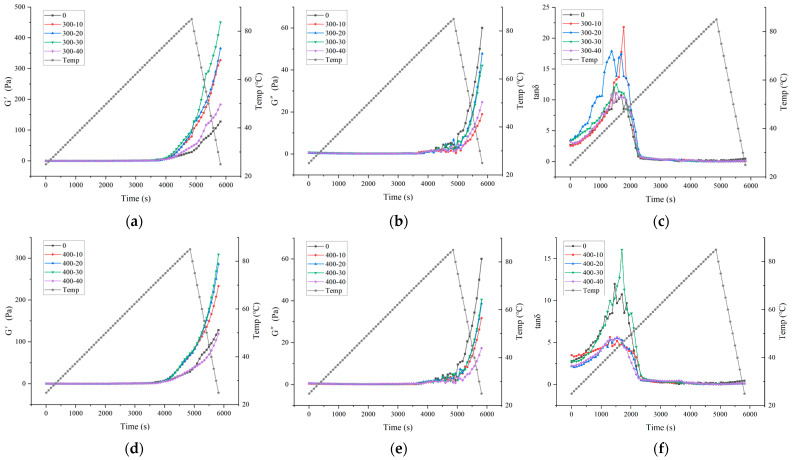
Thermal coagulation temperature gradient curves of bovine plasma protein-carboxymethyl cellulose composite gel at different ultrasonic power and time during heating and cooling. (**a**): Energy storage modulus at 300 W ultrasonic power. (**b**): Loss modulus at ultrasonic power 300 W. (**c**): Loss tangent at ultrasonic power 300 W. (**d**): Energy storage modulus at 400 W ultrasonic power. (**e**): Loss modulus at ultrasonic power 400 W. (**f**): Loss tangent at ultrasonic power 400 W. (**g**): Energy storage modulus at 500 W ultrasonic power. (**h**): Loss modulus at ultrasonic power 500 W. (**i**): Loss tangent at ultrasonic power 500 W. (**j**): Energy storage modulus at 600 W ultrasonic power. (**k**): Loss modulus at ultrasonic power 600 W. (**l**): Loss tangent at ultrasonic power 600 W. (**m**): Energy storage modulus at 700 W ultrasonic power. (**n**): Loss modulus at ultrasonic power 700 W. (**o**): Loss tangent at ultrasonic power 700 W.

**Figure 3 foods-13-00732-f003:**
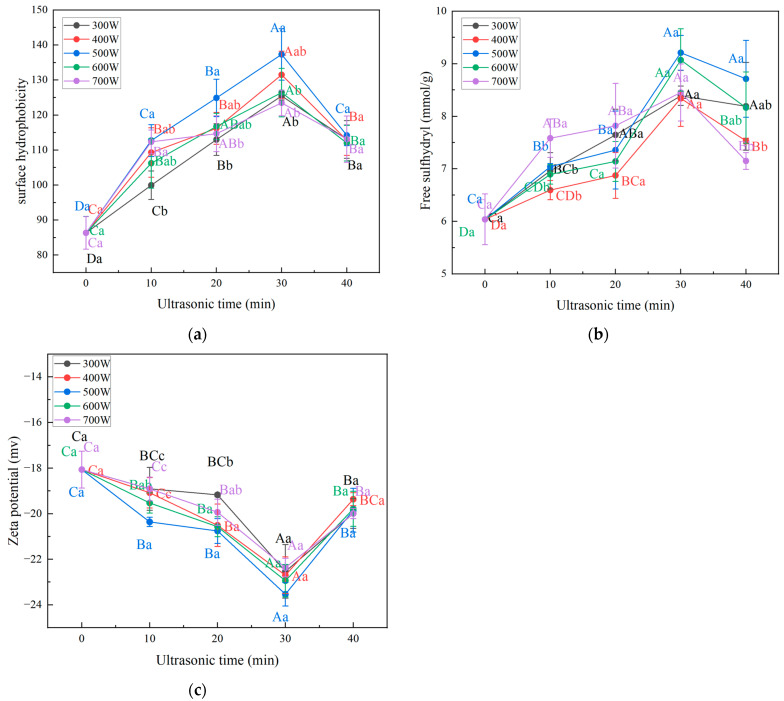
(**a**): Surface hydrophobicity of bovine plasma protein-carboxymethyl cellulose composite gel under different ultrasonic time and power. (**b**): Free sulfhydryl group of bovine plasma protein-carboxymethyl cellulose composite gel under different ultrasonic time and power. (**c**): zeta potential of bovine plasma protein-carboxymethyl cellulose composite gel under different ultrasonic time and power. Lower case letters indicate significant differences between different ultrasound powers. Upper case letters indicate significant differences for different ultrasound times.

**Figure 4 foods-13-00732-f004:**
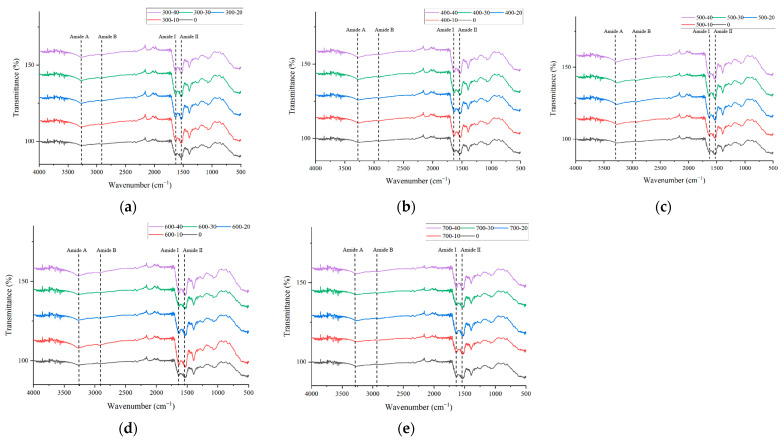
Effect of ultrasonic power and time on infrared spectrum of bovine plasma protein-carboxymethyl cellulose composite gel. (**a**): Ultrasonic power 300 W. (**b**): Ultrasonic power 400 W. (**c**): Ultrasonic power 500 W. (**d**): Ultrasonic power 600 W. (**e**): Ultrasonic power 700 W.

**Figure 5 foods-13-00732-f005:**
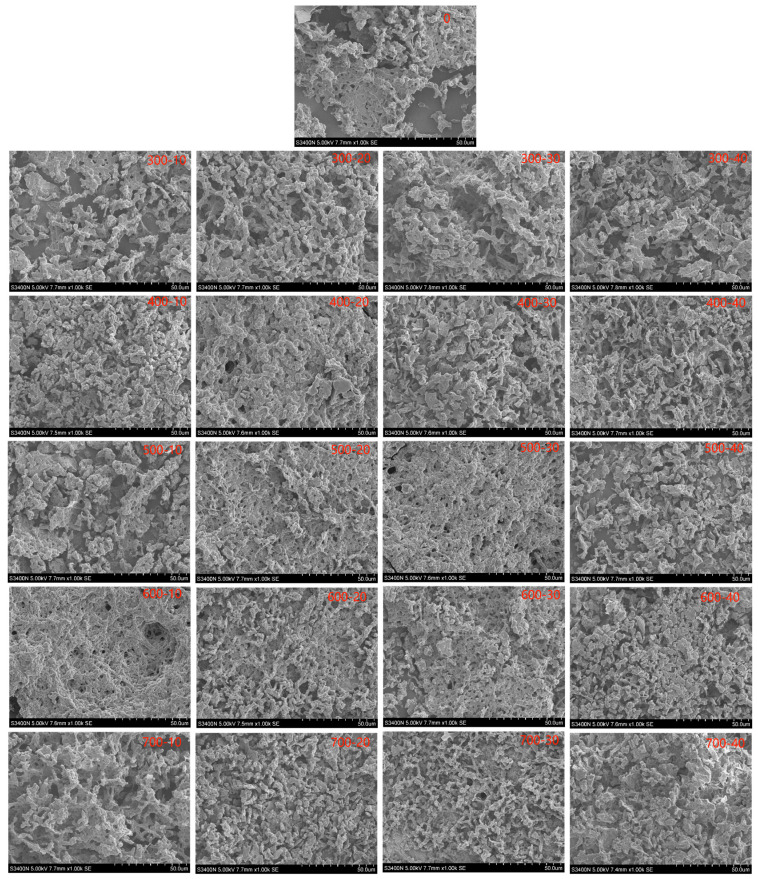
Effect of ultrasonic power and time on microstructure of bovine plasma protein-carboxymethyl cellulose composite gel.

**Table 1 foods-13-00732-t001:** Peak temperature (Tp) and denaturation enthalpy (ΔH) of bovine plasma protein- Carboxymethyl cellulose composite gel at different ultrasonic time and power.

P(W)	Time				
	0 min	10 min	20 min	30 min	40 min
ΔH					
300	109.1040 ^Ba^ ± 13.1167	121.0167 ^Bb^ ± 6.4005	145.0300 ^Ab^ ± 14.2598	154.4000 ^Ac^ ± 9.9226	117.7367 ^Bc^ ± 9.2251
400	109.1040 ^Ba^ ± 13.1167	162.7667 ^Aa^ ± 17.7004	170.1633 ^Aa^ ± 15.8720	173.3300 ^Abc^ ± 12.3669	146.7467 ^Aa^ ± 11.2907
500	109.1040 ^Da^ ± 13.1167	157.9167 ^BCa^ ± 8.6188	179.6900 ^ABa^ ± 12.3100	203.8200 ^Aa^ ± 15.9369	141.6900 ^Cab^ ± 16.3622
600	109.1040 ^Ca^ ± 13.1167	155.3033 ^Ba^ ± 16.9763	174.4033 ^ABa^ ± 8.1869	190.7000 ^Aab^ ± 17.0903	158.4067 ^Ba^ ± 15.2379
700	109.1040 ^Ca^ ± 13.1167	121.1163 ^BCb^ ± 10.6894	142.4567 ^Bb^ ± 9.9826	189.3800 ^Aab^ ± 13.2137	140.0000 ^Bab^ ± 11.2198
Tp					
300	111.2157 ^Ca^ ± 2.4397	115.0693 B^Ca^ ± 4.0570	122.2607 ^Aa^ ± 3.8768	119.0090 ^ABa^ ± 2.0110	111.4880 ^Cc^ ± 3.5947
400	111.2157 ^Ba^ ± 2.4397	113.6807 ^ABa^ ± 2.9780	116.2740 ^ABa^ ± 4.9646	120.2440 ^Aa^ ± 3.7830	119.7403 ^Ab^ ± 2.9010
500	111.2157 ^Ba^ ± 2.4397	119.6007 ^Aa^ ± 2.3291	118.0637 ^Aa^ ± 4.8174	121.6027 ^Aa^ ± 2.7969	118.9860 ^Ab^ ± 3.2742
600	111.2157 ^Ca^ ± 2.4397	119.6127 ^Ba^ ± 3.9875	119.6970 ^Ba^ ± 2.7124	115.7713 ^BCa^ ± 4.2765	127.5853 ^Aa^ ± 3.8497
700	111.2157 ^Ba^ ± 2.4397	113.0253 ^ABa^ ± 4.9895	118.5247 ^Aa^ ± 2.1534	119.5967 ^Aa^ ± 3.5815	117.6173 ^ABb^ ± 3.0144

Note: Data are expressed as the mean ± standard deviation (SD) from triplicate experiments. Different letters (A–C; a–c) in the same column indicate significant differences (*p* < 0.05). Lowercase letters indicate significant differences between different ultrasound powers. Upper case letters indicate significant differences for different ultrasound times.

**Table 2 foods-13-00732-t002:** Effects of ultrasonic power and time on conjugated bimodal ratio of I850/I830 double bands and mole fraction of exposed (hidden) tyrosine residues of bovine plasma protein-carboxymethyl cellulose complex gel.

P(W)	Time				
	0 min	10 min	20 min	30 min	40 min
I850/I830					
300	0.9271 ^Ea^ ± 0.0014	0.9519 ^Dd^ ± 0.0004	0.9725 ^Cd^ ± 0.0010	0.9923 ^Ae^ ± 0.0001	0.9763 ^Be^ ± 0.0007
400	0.9271 ^Ea^ ± 0.0014	0.9754 ^Db^ ± 0.0001	0.9878 ^Bb^ ± 0.0006	1.0003 ^Ac^ ± 0.0002	0.9814 ^Cd^ ± 0.0002
500	0.9271 ^Ea^ ± 0.0014	0.9985 ^Da^ ± 0.0002	1.0091 ^Ba^ ± 0.0003	1.0127 ^Aa^ ± 0.0003	1.0023 ^Ca^ ± 0.0012
600	0.9271 ^Ea^ ± 0.0014	0.9768 ^Db^ ± 0.0020	0.988 ^Bb^ ± 0.0001	1.0015 ^Ab^ ± 0.0003	0.9856 ^Cc^ ± 0.0004
700	0.9271 ^Ea^ ± 0.0014	0.9613 ^Dc^ ± 0.0003	0.9784 ^Cd^ ± 0.0003	0.9987 ^Ad^ ± 0.0003	0.9880 ^Bb^ ± 0.0010
Nexposed					
300	0.5695 ^Ea^ ± 0.0018	0.6025 ^Dd^ ± 0.0005	0.6300 ^Cd^ ± 0.0014	0.6564 ^Ae^ ± 0.0002	0.6351 ^Be^ ± 0.0010
400	0.5695 ^Ea^ ± 0.0018	0.6339 ^Db^ ± 0.0001	0.6504 ^Bb^ ± 0.0008	0.6671 ^Ac^ ± 0.0003	0.6419 ^Cd^ ± 0.0002
500	0.5695 ^Ea^ ± 0.0018	0.6646 ^Da^ ± 0.0003	0.6788 ^Ba^ ± 0.0004	0.6836 ^Aa^ ± 0.0004	0.6697 ^Ca^ ± 0.0015
600	0.5695 ^Ea^ ± 0.0018	0.6357 ^Db^ ± 0.0027	0.6515 ^Bb^ ± 0.0001	0.6686 ^Ab^ ± 0.0004	0.6474 ^Cc^ ± 0.0005
700	0.5695 ^Ea^ ± 0.0018	0.6150 ^Dc^ ± 0.0004	0.6379 ^Bc^ ± 0.0004	0.6649 ^Ad^ ± 0.0004	0.6507 ^Cb^ ± 0.0013
Nburied					
300	0.4305 ^Aa^ ± 0.0018	0.3975 ^Ba^ ± 0.0005	0.3670 ^Ca^ ± 0.0014	0.3436 ^Ea^ ± 0.0002	0.3649 ^Da^ ± 0.0010
400	0.4305 ^Aa^ ± 0.0018	0.3661 ^Bc^ ± 0.0001	0.3496 ^Dc^ ± 0.0008	0.3329 ^Ec^ ± 0.0003	0.3581 ^Cb^ ± 0.0002
500	0.4305 ^Aa^ ± 0.0018	0.3354 ^Bd^ ± 0.0003	0.3212 ^Dd^ ± 0.0004	0.3164 ^Ee^ ± 0.0004	0.3303 ^Ce^ ± 0.0015
600	0.4305 ^Aa^ ± 0.0018	0.3643 ^Bc^ ± 0.0027	0.3485 ^Dc^ ± 0.0001	0.3314 ^Ed^ ± 0.0004	0.3526 ^Cc^ ± 0.0005
700	0.4305 ^Aa^ ± 0.0018	0.3850 ^Bb^ ± 0.0004	0.3621 ^Cb^ ± 0.0004	0.3351 ^Eb^ ± 0.0004	0.3493 ^Dd^ ± 0.0013

Note: Data are expressed as the mean ± standard deviation (SD) from triplicate experiments. Different letters (A–E; a–e) in the same column indicate significant differences (*p* < 0.05). Lowercase letters indicate significant differences between different ultrasound powers. Upper case letters indicate significant differences for different ultrasound times.

**Table 3 foods-13-00732-t003:** Effect of ultrasonic power and time on the secondary structure content of bovine plasma protein-carboxymethyl cellulose complex gel.

P(W)	Time				
	0 min	10 min	20 min	30 min	40 min
β-sheet (%)					
300	31.2616 ^Da^ ± 0.2068	37.4694 ^Cc^ ± 0.2920	39.0639 ^Aa^ ± 0.1086	38.1437 ^Bc^ ± 0.2103	38.9964 ^Aa^ ± 0.1336
400	31.2616 ^Ca^ ± 0.2069	38.0276 ^Bb^ ± 0.1079	38.1562 ^Bb^ ± 0.4473	39.3194 ^Aa^ ± 0.0604	38.2275 ^Bb^ ± 0.1566
500	31.2616 ^Da^ ± 0.2070	37.8261 ^Cb^ ± 0.0896	38.4804 ^Bab^ ± 0.1864	39.3735 ^Aa^ ± 0.1372	38.7028 ^Ba^ ± 0.1684
600	31.2616 ^Ca^ ± 0.2071	38.0805 ^Bb^ ± 0.1330	39.0577 ^Aa^ ± 0.0517	39.1209 ^Aab^ ± 0.1217	37.8202 ^Bc^ ± 0.3425
700	31.2616 ^Da^ ± 0.2072	38.6731 ^ABa^ ± 0.1991	38.3790 ^Bb^ ± 0.5829	38.9611 ^Ab^ ± 0.1383	37.5088 ^Cc^ ± 0.0413
α-helix (%)					
300	12.5224 ^Ca^ ± 0.1144	15.1509 ^Bc^ ± 0.2082	15.6251 ^Aa^ ± 0.0671	15.2771 ^Bb^ ± 0.0549	15.5835 ^Aab^ ± 0.0587
400	12.5224 ^Ca^ ± 0.1145	15.3317 ^Bc^ ± 0.1629	15.5860 ^ABa^ ± 0.4584	15.9235 ^Aa^ ± 0.0070	15.2475 ^Bab^ ± 0.0904
500	12.5224 ^Ca^ ± 0.1146	14.8328 ^Bd^ ± 0.1978	15.0879 ^Bc^ ± 0.0728	15.5864 ^Aab^ ± 0.0099	15.8799 ^Aa^ ± 0.5594
600	12.5224 ^Ca^ ± 0.1147	17.9642 ^Aa^ ± 0.1473	15.7466 ^Ba^ ± 0.1101	15.4468 ^Bb^ ± 0.1478	15.9029 ^Ba^ ± 0.6024
700	12.5224 ^Ca^ ± 0.1148	15.6407 ^Ab^ ± 0.0743	15.4634 ^Aab^ ± 0.2475	15.3586 ^Ab^ ± 0.3997	14.9461 ^Bb^ ± 0.0586
random coiling (%)					
300	39.5772 ^Aa^ ± 0.3574	28.5526 ^Ba^ ± 0.2680	25.5755 ^Dc^ ± 0.2769	27.8534 ^Ca^ ± 0.1366	25.6389 ^Dc^ ± 0.1423
400	39.5772 ^Aa^ ± 0.3575	27.9763 ^Bb^ ± 0.0850	26.5534 ^Cb^ ± 0.2530	22.9586 ^Dcd^ ± 0.1113	27.7542 ^Bab^ ± 0.0140
500	39.5772 ^Aa^ ± 0.3576	27.9113 ^Bb^ ± 0.0137	27.6297 ^Ba^ ± 0.1347	21.5784 ^Dd^ ± 0.0452	27.2294 ^Cb^ ± 0.2384
600	39.5772 ^Aa^ ± 0.3577	25.3025 ^Cc^ ± 0.2499	23.0848 ^Dd^ ± 0.1621	25.3856 ^Cb^ ± 0.2657	27.3311 ^Bb^ ± 0.6949
700	39.5772 ^Aa^ ± 0.3578	22.0249 ^Cd^ ± 0.1181	23.7199 ^Cd^ ± 0.8115	23.7118 ^Cbc^ ± 2.2435	28.0864 ^Ba^ ± 0.0881
β-turn (%)					
300	16.6389 ^Ca^ ± 0.3990	18.8272 ^Bc^ ± 0.5783	19.7355 ^Ab^ ± 0.1239	18.7257 ^Bd^ ± 0.1208	19.7811 ^Aa^ ± 0.1812
400	16.6389 ^Da^ ± 0.3991	18.6644 ^Cc^ ± 0.1403	19.7043 ^Bb^ ± 0.2395	21.7985 ^Ab^ ± 0.0578	18.7708 ^Cc^ ± 0.1495
500	16.6389 ^Ea^ ± 0.3992	19.4298 ^Bb^ ± 0.1051	18.8019 ^Cc^ ± 0.1213	23.4618 ^Aa^ ± 0.1327	18.1879 ^Dd^ ± 0.4287
600	16.6389 ^Da^ ± 0.3993	18.6529 ^Cc^ ± 0.2308	22.1109 ^Aa^ ± 0.0989	20.0467 ^Bc^ ± 0.1804	18.9459 ^Cbc^ ± 0.4595
700	16.6389 ^Da^ ± 0.3994	23.6614 ^Aa^ ± 0.0813	21.8556 ^Ba^ ± 0.6161	22.5506 ^ABab^ ± 1.5222	19.4588 ^Cab^ ± 0.0449

Note: Data are expressed as the mean ± standard deviation (SD) from triplicate experiments. Different letters (A–E; a–c) in the same column indicate significant differences (*p* < 0.05). Lowercase letters indicate significant differences between different ultrasound powers. Upper case letters indicate significant differences for different ultrasound times.

## Data Availability

The original contributions presented in the study are included in the article, further inquiries can be directed to the corresponding author.

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
