# Peer review of "Effect of Ultrasonic Treatment on the Physicochemical Properties of Bovine Plasma Protein-Carboxymethyl Cellulose Composite Gel"

_foods, 2024, doi:10.3390/foods13050732_

Round 1

Reviewer 1 Report

Comments and Suggestions for Authors

This study is original well organized and writted. The authors woith various characterization and experimental methods have studied in depth the effect of ultrasonic treatment on the physicochemical properties of bovine plasma protein-carboxymethyl cellulose composite gel. 

Althought some revisions must be done before farther evaliuation.

Best wishes!

Author Response

Point 1: The absttract need to be more atractive for readers. Start with a more general sentence such as what is the general problem this study wants to resolve or under what kind of global researh trend is this study.... Also and numerical results in the abstract.

Response 1: Thank you for your careful reading of our manuscript. Based on your comments we have modified the first sentence of the abstract to read: in order to improve the stability of bovine plasma protein-carboxymethylcellulose composite gels and to expand the utilization of animal by-product resources. Done as requested, please check line 9, 10.

Point 2: Introduction seems good but the innovative points of current study against other ones must be highlighted better in the end of introduction section.

Response 2: Thank you for your careful reading of our manuscript. According to your comments, we have changed the last sentence in the introduction to read: In this study, we investigated the mechanism of ultrasound action on protein-polysaccharide composite gels to provide a theoretical basis for the development and application of bovine plasma protein-carboxymethylcellulose composite gels in the field of novel foods. Done as requested, please check line 95 - 98.

Point 3:: Line 3: WHC and number in equation.

Response 3: Thank you for your valuable advice. We have made changes to the formatting of the letters and numbers in the formula. Done as requested, please check line 142.

Point 4: Statistical analysis, and the results of statistical analysis in supplamentary file...

Response 4: Thank you for your careful reading of our manuscript. We have changed the paragraph on statistical analysis to read: statistical analysis was performed using SPSS 18 data analysis software (IBM, New York, USA). Data were expressed as mean ± standard deviation. One-way analysis of variance (ANOVA) was used for the significance of main effects, and Duncan's multiple extreme variance test (p<0.05) was used for significant differences between groups. Two-factor ANOVA was used to analyze the interaction between factors. p < 0.05 was considered statistically significant for differences between models. All graphs were plotted using Origin 8.5 software (Origin Lab Corp., MA, USA). All tests were repeated three times. Done as requested, with changes marked in red in the manuscript. Thanks for your correction.

Please find attached the manuscript of the paper.

Reviewer 2 Report

Comments and Suggestions for Authors

The manuscript deals with an interesting topic of investigating the influence of ultrasonic treatment and ultrasonic time on the structure and gel properties of bovine plasma protein-carboxymethyl cellulose composite gel.

Below, I offer some minor remarks which should be addressed:

Line 12:  Change “ultrasound durations”  to “ultrasound times”.

Line 14: Please, explain what is “WHC”.

Line 20: Change “duration” to “ time”.

Line 106: (Henan, China).

Line 107: grade.

Line 115: There are many space mistakes, 85 °C, 4 °C.

Line 112: Please, add the brand and model.

Line 119: Please, add the brand and model.

Material and methods: Please, put the information in the past.

Line 130: Rewrite this sentence.

Line 137: Please, add the brand and model.

Line 155: Please, add the brand and model.

Line 186: Please, add the brand and model.

Line 270: Zhao et al. [15].

Line 297:  Reference [41] is not “Rassoul M et al”.

3.5.2. Change to “Free sulfhydryl content”.

3.5.3. Change to “Zeta potential”.

Line 374:  Please, explain what is “HIU”.

Line 395: Reference [54] is not “Ning et al”.

Please, review all references.

Comments on the Quality of English Language

Minor editing of English language required.

Author Response

Point 1: Line 12:  Change “ultrasound durations”  to “ultrasound times”.

Response 1: We apologize for our careless mistake and thank you for the reminder. Based on your comments, “ultrasound durations” has been changed to “ultrasound times”. This study investigated the impact of different ultrasound powers (300, 400, 500, 600, and 700 W) and ultrasound times (0, 10, 20, 30, and 40 min) on the functional properties. Done as requested, please check line 11.

Point 2: Line 14: Please, explain what is “WHC”.

Response 2: Thank you for your careful reading of our manuscript. Based on your comments, WHC has been explained here. The results showed that the gel strength, water holding capacity (WHC) and thermal stability of the composite gel were the highest in all treatment conditions at 500 W and 30 min. Done as requested, please check line 14.

Point 3: Line 20: Change “duration” to “ time”.

Response 3: We apologize for our careless mistake and thank you for the reminder. Based on your comments, “duration” has been changed to “time”. In conclusion, appropriate ultrasonic power and time can significantly improve the functional and structural properties of the composite gel. Done as requested, please check line 19.

Point 4: Line 106: (Henan, China).

Response 4: Thank you for your careful reading of our manuscript and for the reminder. We have checked the manuscript for formal error. Done as requested, with changes marked in red in the manuscript. Thanks for your correction, please check line 108.

Point 5: Line 107: grade.

Response 5: Thank you for pointing out this problem. We have graded the titles here, please check line 113.

Point 6: Line 115: There are many space mistakes, 85 °C, 4 °C.

Response 6: Thank you for your careful reading of our manuscript and for the reminder. We have checked the manuscript for formal error. Done as requested, with changes marked in red in the manuscript. Thanks for your correction, please check line 124.

Point 7: Line 112: Please, add the brand and model.

Response 7: Thank you for your valuable advice. We have added the make and model of the instrumentation to the manuscript. The revised section has been marked in red in the manuscript.

Point 8: Line 119: Please, add the brand and model.

Response 8: Thank you for your valuable advice. We have added the make and model of the instrumentation to the manuscript. The revised section has been marked in red in the manuscript.

Point 9: Material and methods: Please, put the information in the past.

Response 9: Thank you for your suggestion. We have changed the tense in Materials and Methods to the past tense. The revised section has been marked in red in the manuscript.

Point 10: Line 130: Rewrite this sentence.

Response 10: Thank you for pointing out the problem. We have changed this statement to: accurately measure the weight of the tube containing the gel sample before and after centrifugation and calculate the composite gel water holding capacity (WHC) according to the following equation. The revised section has been marked in red in the manuscript. Please check line 139-142.

Point 11: Line 137: Please, add the brand and model.

Response 11: Thank you for your valuable advice. We have added the make and model of the instrumentation to the manuscript. The revised section has been marked in red in the manuscript.

Point 12: Line 155: Please, add the brand and model.

Response 12: Thank you for your careful reading of our manuscript and for the reminder. We have added the make and model of the instrumentation to the manuscript. The revised section has been marked in red in the manuscript. Please check line 166.

Point 13: Line 186: Please, add the brand and model.

Response 13: Thank you for your valuable advice. We have added the make and model of the instrumentation to the manuscript. The revised section has been marked in red in the manuscript. Please check line 198.

Point 14: Line 270: Zhao et al. [15].

Response 14: Thank you for your careful reading of our manuscript and for the reminder. We have checked the manuscript for formal error. Done as requested, with changes marked in red in the manuscript. Thanks for your correction, please check line 290.

Point 15: Line 297:  Reference [41] is not “Rassoul M et al”.

Response 15: Thank you for your careful reading of our manuscript and for the reminder. We have checked and corrected the writing error here. Done as requested, with changes marked in red in the manuscript. Thanks for your correction, please check line 322.

Point 16: 3.5.2. Change to “Free sulfhydryl content”.

Response 11: We apologize for our careless mistake and thank you for the reminder. Based on your comments, “Surface hydrophobicity” has been changed to “Free sulfhydryl content”. Done as requested, please check line 361.

Point 17: 3.5.3. Change to “Zeta potential”.

Response 17: We apologize for our careless mistake and thank you for the reminder. Based on your comments, “Zeta potentia” has been changed to “Zeta potential”. Done as requested, please check line 380.

Point 18: Line 374:  Please, explain what is “HIU”.

Response 18: Thank you for your careful reading of our manuscript and for the reminder. Based on your comments, “HIU” has been changed to “high intensity ultrasound”. Done as requested, please check line 401.

.

Point 19: Line 395: Reference [54] is not “Ning et al”.

Response 19: Thank you for your careful reading of our manuscript and for the reminder. We have checked and corrected the writing error here. Done as requested, with changes marked in red in the manuscript. Thanks for your correction, please check line 429.

Point 20: Please, review all references.

Response 20: Thank you for your careful reading of our manuscript and for the reminder. We have checked all references cited in the text. Done as requested, with changes marked in red in the manuscript. Thanks for your correction.

Please find attached the manuscript of the paper.

Reviewer 3 Report

Comments and Suggestions for Authors

Manuscript Number: foods- 2863510 -peer-review-v1

Title: Effect of ultrasonic treatment on the physicochemical properties of bovine plasma protein-carboxymethyl cellulose composite gel

Liyuan Wang, Yu Ma; Ruheng Shen, Li Zhang, Long He, Yuling Qu, Xiaotong Ma, Guoyuan Ma, Zhaobin Guo, Cheng Chen, Hongbo Li and Xiangying kong

Overview and general recommendation:

The manuscript is interesting due to the use of many modern research methods that allow to explain effect of ultrasonic treatment on the physicochemical properties of bovine plasma protein-carboxymethyl cellulose composite gel. However, the manuscript contains many shortcomings and ambiguities and needs to be thoroughly revised.

Major comments:

English language - I suggest that the manuscript be read by a native speaker, as the English language needs improvement. Additionally, please change the tense used in the text to the past tense, e.g. in lines: 109, 101, 114.

What was the effect of the temperature of the solution after sonication? Please complete the information on the temperature of the samples after US treatment?

The statistics are misinterpreted, as if the authors were only guided by the results of the tests and not by what the statistical analysis indicates. Text needs to be analyzed again and corrected according to the results of the statistical analysis.

Minor comments:

Line 21 – „The aim of this study was…” - the aim in the abstract should not come at the end, after the results and conclusions have been discussed.

Line 23 – “ideal texture” - What does ideal texture mean? Is this term correct?

Lines 41-42 – “ Hence, the combination of bovine blood plasma protein with polysaccharides is essential to improve its surface activity and gelation properties.- please provide references to this sentence.

Line 104 – “Bovine plasma protein powder (BPP)” - What was the protein content in the obtained powder?

Material - Please add all rests, buffers and solutions that were used during the tests, not only to prepare the mixtures. E.g. PBS buffer (add also pH), KBr

Lines 109- 110 – “Take a certain amount of distilled water, add 7% bovine plasma protein and 0.6% 109 carboxymethyl cellulose, stir well.” - This is not a recipe on how to do it, just information on how such samples were made, so it needs to be modified to: Planned 7% bovine plasma protein and 0.6% carboxymethylcellulose were added to a certain amount of distilled water and stired well.

Methods - add equipment names - homogenizer, magnetic stirrer, ultrasonic processor, equipment for heat treatment (line114), instrumental texture analyzer, dynamic rheometer, DSC instrument, Flourescence Spectrofluorometer, homogenizer and centrifuge (line 158), Microscope (line 187)

Lines 118-119 – “Gel strength was measured using a described method previously with slight modifications [18].” - The methodology must be described precisely, including the principle of the method, equipment and parameters (very important is temperature), and existing publications may be referred to. However, this manuscript must constitute a separate whole, it must be understandable to the reader regardless of whether the research method has been published before (the exception are standard methods included in international standards, where it is enough to quote only the regulation number).

Line 122 – please add also the name of software used.

Line 128 – “at 8000 R /min for 10 min.” – pls add the temperature.

Line 162 – “complete reaction of the mixture.” – What kind of the reaction?

2.10. Statistical analysis - What was the method that was used for statistical analysis? This is especially important because the authors examine two variable factors - one is the power and the other is the sonication time? For example, it is done differently in table 1, please correct the entire statistic and additionally describe what method/test this statistic was calculated. How many replies of the experiment were provided?

Line 198 – “texture of gelatin products” - I guess you meant gels and not gelatin?

Line 201 – “Gel strength” - Force, according to SI units, should be expressed in N.

Lines 201- 202” When the ultrasound time was 30 minutes and the ultrasound power was 500 W, the maximum gel strength of the composite gel was 297.6 g.” - The charts are illegible. It is not known whether the differences between solutions are statistically significant (e.g. 400W and 500W).

Lines 203-206 - The English language needs improvement because this sentence is incomprehensible.

Line 212 – “lower than that of other groups” It is not known what these other groups were, not what the control group was, so this needs to be clarified so that the results can be compared with those presented in the authors' research.

Figures 1a, 1b, 3a, 3b, 3c - The charts are illegible and the selected statistics are incomprehensible. The graphs should be enlarged and the statistics of the impact of changing ultrasound time at a given ultrasound power should be marked on the graphs (e.g. in lowercase letters) and, separately, the impact of changing ultrasound power at a given ultrasound time (e.g. in capital letters). After performing such an analysis, the description of the results should be reworded taking into account the results of the statistical analysis.

Figure 1b - units missing on WHC axis [%].

Line 222 – “heat-induced protein gel” - it is not understood whether the authors' goal was to compare the ultrasonically obtained gels to thermally induced gels? Because this is the second time they have made such a comparison?

Line 228 – “water-holding capacity of the composite gel was the strongest, increasing by 27.39% compared” - Please calculate it again, because in my opinion the difference is much smaller? around 14.

Line 290 – “were increased” not all, 400W - 0 - 10 -20 min are statistically the same?

Lin 291 – “…stability of composite gels.” - It should be emphasized here that extending the time from 30 to 40 min (500, 600, 700 W) reduces delta H.

Line 293 – “reaches the highest.” Is it the same as 600 and 700?

Tables 1,2,3 – “different letters (A–C; a–c) in the same column indicate significant differences (P < 0.05” - This may be misleading, you have to write A-C in row and a-c in column.

                 The lowercase letters indicate that different ultrasonic time have significant differences. The capital letters indicate that different ultrasonic power have significant differences.” - And not the other way around? if the capital letters refer to the influence of power - then at 0 operation time (i.e. no ultrasonic operation) the values (the same) differ from each other???

Line 313 – “increases and then decreases.” - Please indicate when this value starts to decrease.

Lines 319-320 – “When the ultrasound time is too long and the ultrasound power is too strong, the surface hydrophobicity of the protein decreases” - The effect of ultrasound size cannot be determined because the graphs are too small to read this information.

Figure 3c - If the values on the y-axis are negative, then the graph should probably be the other way round?

Line 334 - 3.5.2. Surface hydrophobicity - Isn't this title the same as the previous chapter?

Lines 338-339 – “As shown in Figure 3. (b), with the increase in ultrasound power, the content of free sulfhydryl in the protein also increases” - Only at the beginning, because after 30 minutes it is different, e.g. for 700W.

Lines 363-364 – “Furthermore, the absolute values of the Zeta potential for all self-assembled samples are greater than 20 mV.” - It's not clear what samples authors mean? some of the indicated values are below 20, for absolute value.

Lines 389-390 – “decreasing with increasing ultrasound power and time.” - this is not entirely true - because according to statistics for 400W it increases from 20 to 30 minutes, similarly there are not much differences for shorter times - for 400 and 600W - the whole thing therefore requires re-analysis and redescription,

Table 2 the table lists the results of Nexposed and Nburied, which are not described in the text, maybe they are unnecessary?

Line 429 – “significantly higher than those in the control group- And here again it is not consistent with the statistical analysis - taking into account 300W and times 0, 20 and 40 min, please analyze the description of the results again in terms of compliance with the statistical analysis.

Table 3 - is the title good? Isn't this the title from the previous table?

Conclusion - please modify your conclusions in accordance with the updated version of the results and discussion, which will be based on statistical analysis

Comments on the Quality of English Language

English language - I suggest that the manuscript be read by a native speaker, as the English language needs improvement. Additionally, please change the tense used in the text to the past tense, e.g. in lines: 109, 101, 114.

Author Response

Please find attached the manuscript of the paper.

Point 1: English language - I suggest that the manuscript be read by a native speaker, as the English language needs improvement. Additionally, please change the tense used in the text to the past tense, e.g. in lines: 109, 101, 114.

Response 1: Thank you for your careful reading of our manuscript and for the reminder. We have changed the tense in Materials and Methods to the past tense. The revised section has been marked in red in the manuscript.

Point 2: What was the effect of the temperature of the solution after sonication? Please complete the information on the temperature of the samples after US treatment?

Response 2: Thank you for your careful reading of our manuscript. We have added information about the temperature during sonication to the manuscript, and the samples were kept in an ice water bath at all times during sonication to ensure that the temperature of the samples was kept below 25 °C. please check line 121.

Point 3: The statistics are misinterpreted, as if the authors were only guided by the results of the tests and not by what the statistical analysis indicates. Text needs to be analyzed again and corrected according to the results of the statistical analysis.

Response 3: Thank you for your careful reading of our manuscript. We have been examining the analysis of the results of the manuscript and have made corrections based on the results of the statistical analysis, which are highlighted in red in the manuscript.

Point 4: Line 21 – „The aim of this study was…” - the aim in the abstract should not come at the end, after the results and conclusions have been discussed.

Response 4: Thank you very much for pointing out this important issue and we agree with your comments. We have changed this sentence to read: It was found that controlling the thermal aggregation behavior of composite gels by adjusting the ultrasonic power and time is an effective strategy to enable the optimization of composite gel texture and water retention properties, please check line 20-23.

Point 5: Line 23 – “ideal texture” - What does ideal texture mean? Is this term correct?

Response 5: Thank you for your careful reading of our manuscript and for the reminder. Based on your comments, “ideal texture” has been changed to “optimization of composite gel texture”. Done as requested, please check line 22.

Point 6: Lines 41-42 – “ Hence, the combination of bovine blood plasma protein with polysaccharides is essential to improve its surface activity and gelation properties.” - please provide references to this sentence.

Response 6: Thank you for your careful reading of our manuscript and for the reminder. We have added references in the manuscript. Done as requested, please check line 42.

Point 7: Line 104 – “Bovine plasma protein powder (BPP)” - What was the protein content in the obtained powder?

Response 7: Thank you for your valuable advice. We have added the base ingredient of Bovine Plasma Protein Powder to our manuscript which contains information about its protein content. Done as requested, please check line 106, 107.

Point 8: Material - Please add all rests, buffers and solutions that were used during the tests, not only to prepare the mixtures. E.g. PBS buffer (add also pH), KBr.

Response 8: Thank you for your valuable advice. We have added information about all the buffers used during testing as well as KBr in the manuscript. Done as requested, please check line 108-112.

Point 9: Lines 109- 110 – “Take a certain amount of distilled water, add 7% bovine plasma protein and 0.6% 109 carboxymethyl cellulose, stir well.” - This is not a recipe on how to do it, just information on how such samples were made, so it needs to be modified to: Planned 7% bovine plasma protein and 0.6% carboxymethylcellulose were added to a certain amount of distilled water and stired well.

Response 9: Thank you for your suggestion. We have changed this to 7% bovine plasma protein and 0.6% carboxymethylcellulose added to a quantity of distilled water and stirred well. Done as requested, please check line 114, 115.

Point 10: Methods - add equipment names - homogenizer, magnetic stirrer, ultrasonic processor, equipment for heat treatment (line114), instrumental texture analyzer, dynamic rheometer, DSC instrument, Flourescence Spectrofluorometer, homogenizer and centrifuge (line 158), Microscope (line 187).

Response 10: Thank you for your careful reading of our manuscript and for the reminder. We have added the name and model of the device to the manuscript.

Point 11: Lines 118-119 – “Gel strength was measured using a described method previously with slight modifications [18].” - The methodology must be described precisely, including the principle of the method, equipment and parameters (very important is temperature), and existing publications may be referred to. However, this manuscript must constitute a separate whole, it must be understandable to the reader regardless of whether the research method has been published before (the exception are standard methods included in international standards, where it is enough to quote only the regulation number).

Response 11: Thank you for your careful reading of our manuscript and we agree with you. We have rewritten the gel strength test method in the manuscript as follows: gel strength was assessed by the method of Wang et al. Gel strength was assessed using an Instrumental Tissue Analyzer (TA). xT Express, Stable Micro Systems) equipped with a P/0.5R test probe. The test parameters were set as follows: trigger force of 5 g; prediction speed of 1.0 mm/s; test speed of 1.0 mm/s; post-test speed of 1.0 mm/s; and puncture distance of 8.0 mm. Gel strength analysis was performed using the Exponent Connect software that comes with the instrument. Done as requested, please check line 128-133.

Point 12: Line 122 – please add also the name of software used.

Response 12: Thank you for your careful reading of our manuscript and for the reminder. We have added the name of the software in the manuscript as Exponent Connect software. Done as requested, with changes marked in red in the manuscript. Thanks for your correction.

Point 13: Line 128 – “at 8000 R /min for 10 min.” – pls add the temperature.

Response 13: Thank you for your careful reading of our manuscript and thanks for your suggestion. We have added a temperature of 4°C to the manuscript. Done as requested, please check line 137, 138.

Point 14: Line 162 – “complete reaction of the mixture.” – What kind of the reaction?

Response 14: Thank you for reading our manuscript carefully. We believe that the reaction here means that the mixed various reagents are heated with the sample at 40°C for 40 minutes to ensure that they are completely mixed and homogeneous, the reaction is completed, the absorbance of the fully reacted solution at 412 nm is obtained, and the free sulfhydryl content of the sample is analyzed.

Point 15: 2.10. Statistical analysis - What was the method that was used for statistical analysis? This is especially important because the authors examine two variable factors - one is the power and the other is the sonication time? For example, it is done differently in table 1, please correct the entire statistic and additionally describe what method/test this statistic was calculated. How many replies of the experiment were provided?

Response 15: Thank you for your careful reading of our manuscript and for the reminder. We have changed the paragraph on statistical analysis to read: statistical analysis was performed using SPSS 18 data analysis software (IBM, New York, USA). Data were expressed as mean ± standard deviation. One-way analysis of variance (ANOVA) was used for the significance of main effects, and Duncan's multiple extreme variance test (p<0.05) was used for significant differences between groups. Two-factor ANOVA was used to analyze the interaction between factors. p < 0.05 was considered statistically significant for differences between models. All graphs were plotted using Origin 8.5 software (Origin Lab Corp., MA, USA). All tests were repeated three times.. Done as requested, with changes marked in red in the manuscript. Thanks for your correction.

Point 16: Line 198 – “texture of gelatin products” - I guess you meant gels and not gelatin?

Response 16: We apologize for our careless mistake and thank you for the reminder. Based on your comments, “gelatin products” has been changed to “gel products”. Done as requested, please check line 214.

Point 17: Line 201 – “Gel strength” - Force, according to SI units, should be expressed in N.

Response 17: We apologize for our careless mistake and thank you for the correction. We have modified the units of gel strength in Figure 1(a) to N. Done as requested, please check line 232. Figure 1(a) has now been modified as follows:

(a)

(b)

Figure.5 Gel strength of bovine plasma protein-Carboxymethyl cellulose composite gel under different ul-trasonic time and power. (b): WHC of bovine plasma protein-Carboxymethyl cellulose composite gel under different ultrasonic time and power. Different superscript letters mean significant dif-ferences between values with different ultrasound power and ultrasound time (P < 0.05). Lower case letters indicate significant differences between different ultrasound powers. Upper case let-ters indicate significant differences for different ultrasound times. The samples without ultrasonic treatment (ultrasonic time 0 minutes) were the control samples.

Point 18: Lines 201- 202” When the ultrasound time was 30 minutes and the ultrasound power was 500 W, the maximum gel strength of the composite gel was 297.6 g.” - The charts are illegible. It is not known whether the differences between solutions are statistically significant (e.g. 400W and 500W).

Response 18: Thank you for your valuable advice. Based on your comments we have changed this sentence to read that the gel strength of the composite gels with ultrasound times of 30 and 40 minutes were significantly higher than those of the control group (p < 0.05) and that the gel strength of the composite gels reached its maximum at 30 minutes of ultrasound time when the ultrasound power was the same, except for the treatment group with an ultrasound power of 300W. Done as requested, please check line 215-219.

Point 19: Lines 203-206 - The English language needs improvement because this sentence is incomprehensible.

Response 19: Thank you for your suggestion. We have changed this statement to the fact that the increase in the strength of the composite gel may be related to the action of ultrasound, which generates microfluidization and cavitation, and promotes covalent cross-linking between the protein molecules in the composite gel under the action of thermal processing. Done as requested, please check line 219-222.

Point 20: Line 212 – “lower than that of other groups” It is not known what these other groups were, not what the control group was, so this needs to be clarified so that the results can be compared with those presented in the authors' research.

Response 20: Thank you for pointing out the problem. We have stated in the manuscript that the other groups were sonicated gels alone and combined sonicated gels with the addition of konjac glucomannan and we thank you for your guidance.

Point 21: Figures 1a, 1b, 3a, 3b, 3c - The charts are illegible and the selected statistics are incomprehensible. The graphs should be enlarged and the statistics of the impact of changing ultrasound time at a given ultrasound power should be marked on the graphs (e.g. in lowercase letters) and, separately, the impact of changing ultrasound power at a given ultrasound time (e.g. in capital letters). After performing such an analysis, the description of the results should be reworded taking into account the results of the statistical analysis.

Response 21: Thank you for your careful reading of our manuscript and we agree with you. We have labeled the statistics of the data in Figures 1a, 1b, 3a, 3b, and 3c with upper and lower case letters, with lower case letters indicating significant differences between different ultrasound powers. Upper case letters indicate significant differences between different ultrasound times.

Point 22: Figure 1b - units missing on WHC axis [%].

Response 22: We apologize for our careless mistake and thank you for the correction. We have added WHC in % to Figure 1(b). Done as requested, please check line 232.

Point 23: Line 222 – “heat-induced protein gel” - it is not understood whether the authors' goal was to compare the ultrasonically obtained gels to thermally induced gels? Because this is the second time they have made such a comparison?

Response 23: Thank you for your careful reading of our manuscript. We consider all samples to be heat-induced gels because both the unsonicated and sonicated gels were subsequently treated by thermal processing to form gels. In order to avoid ambiguity here, we have replaced "heat-induced protein gel" with "composite gel". Done as requested, please check line 241.

Point 24: Line 228 – “water-holding capacity of the composite gel was the strongest, increasing by 27.39% compared” - Please calculate it again, because in my opinion the difference is much smaller? around 14.

Response 24: Thank you for reading our manuscript carefully and for your valuable advice. What we are trying to convey here is that this group of samples had a 27.39% increase in water holding capacity over the control sample, with 27.39% representing the growth rate.

Point 25: Line 290 – “were increased” not all, 400W - 0 - 10 -20 min are statistically the same?

Response 25: Thank you for your careful reading of our manuscript and for the reminder. We have changed this statement to Tp and ΔH were both elevated in the sonicated composite gels compared to the unsonicated composite gels, and the ΔH values were significantly higher in the samples with sonication times of 20 and 30 min and sonication powers of 400, 500, and 600 W than in the unsonicated samples. Done as requested, with changes marked in red in the manuscript. Thanks for your correction.

Point 26: Lin 291 – “…stability of composite gels.” - It should be emphasized here that extending the time from 30 to 40 min (500, 600, 700 W) reduces delta H.

Response 26: Thank you for your careful reading of our manuscript. We have changed this sentence to ΔH value of the composite gel reaches the maximum value at the same power when the sonication time is 30 min. ΔH value reaches the maximum value of 203.82 J/g when the sonication power is 500 W and the sonication time is 30 min. ΔH value of the composite gel decreases when the sonication time is extended from 30 min to 40 min. The results showed that proper sonication could improve the thermal stability of the composite gels, while too much sonication time would destabilize the protein gels. Done as requested, with changes marked in red in the manuscript. Thanks for your correction.

Point 27: Line 293 – “reaches the highest.” Is it the same as 600 and 700?

Response 27: Thank you for your valuable advice. We have changed this sentence to ΔH value of the composite gel reaches the maximum value at the same power when the sonication time is 30 min. ΔH value reaches the maximum value of 203.82 J/g when the sonication power is 500 W and the sonication time is 30 min. ΔH value of the composite gel decreases when the sonication time is extended from 30 min to 40 min. The revised section has been marked in red in the manuscript.

Point 28: Tables 1,2,3 – “different letters (A–C; a–c) in the same column indicate significant differences (P < 0.05” - This may be misleading, you have to write A-C in row and a-c in column. “The lowercase letters indicate that different ultrasonic time have significant differences. The capital letters indicate that different ultrasonic power have significant differences.” - And not the other way around? if the capital letters refer to the influence of power - then at 0 operation time (i.e. no ultrasonic operation) the values (the same) differ from each other?

Response 28: We apologize for our careless mistake and thank you for the reminder. Based on your comments, the sentence has been changed to read: "Lower case letters indicate significant differences between different ultrasound powers. Upper case letters indicate significant differences for different ultrasound times. ". Done as requested, please check line 330, 438.

Point 29: Line 313 – “increases and then decreases.” - Please indicate when this value starts to decrease.

Response 29: Thank you for your suggestion. We have changed this sentence to read that the surface hydrophobicity first tends to increase with the extension of the sonication time, and the surface hydrophobicity of the composite gel reaches its maximum at 30 minutes of sonication time, and after 30 minutes, the surface hydrophobicity of the composite gel tends to decrease. Done as requested, please check line 337-340.

Point 30: Lines 319-320 – “When the ultrasound time is too long and the ultrasound power is too strong, the surface hydrophobicity of the protein decreases” - The effect of ultrasound size cannot be determined because the graphs are too small to read this information.

Response 30: Thank you for pointing out the problem. We have enlarged all the diagrams in the manuscript so that the reader can get better access to the information.

Point 31: Figure 3c - If the values on the y-axis are negative, then the graph should probably be the other way round?

Response 31: Thank you for your careful reading of our manuscript and we agree with you. We have adjusted the axes of Fig. 3(c) from the manuscript. Done as requested, please check line 359.

Point 32: Line 334 - 3.5.2. Surface hydrophobicity - Isn't this title the same as the previous chapter?

Response 32: We apologize for our careless mistake and thank you for the reminder. Based on your comments, “Surface hydrophobicity” has been changed to “Free sulfhydryl content”. Done as requested, please check line 361.

Point 33: Lines 338-339 – “As shown in Figure 3. (b), with the increase in ultrasound power, the content of free sulfhydryl in the protein also increases” - Only at the beginning, because after 30 minutes it is different, e.g. for 700W.

Response 33: Thank you for your careful reading of our manuscript and thanks for your suggestion. We have checked and modified this sentence in the original manuscript to read as shown in Figure 3. (b) The content of free sulfhydryl groups in proteins increased with increasing sonication power. The free sulfhydryl content in the composite gel increased and then decreased when the ultrasonic time was increased from 0 to 40 minutes and the ultrasonic power was increased from 300 to 700 W. The free sulfhydryl content in the composite gel increased and then decreased. The maximum content of free sulfhydryl groups in the composite gel was 9.21 mmol/g when the ultrasonication time was 30 min and the ultrasonication power was 500W. Done as requested, please check line 365-369.

Point 34: Lines 363-364 – “Furthermore, the absolute values of the Zeta potential for all self-assembled samples are greater than 20 mV.” - It's not clear what samples authors mean? some of the indicated values are below 20, for absolute value.

Response 34: Thank you for reading our manuscript carefully and for your valuable advice. I apologize for my carelessness, we have checked and revised this sentence in the original manuscript to read that the absolute value of zeta potential of the samples with a sonication time of 30 min were all greater than 20 mV. Done as requested, with changes marked in red in the manuscript.

Point 35: Lines 389-390 – “decreasing with increasing ultrasound power and time.” - this is not entirely true - because according to statistics for 400W it increases from 20 to 30 minutes, similarly there are not much differences for shorter times - for 400 and 600W - the whole thing therefore requires re-analysis and redescription, Table 2 the table lists the results of Nexposed and Nburied, which are not described in the text, maybe they are unnecessary?

Response 35: Thank you for your careful reading of our manuscript and for the reminder. We have checked and changed this sentence in the original manuscript to read that it increases and then decreases with increasing ultrasound power and ultrasound time, and reaches a maximum at an ultrasound power of 500 W and an ultrasound time of 30 min. We have checked the data in Table 2 and corrected the meaning of the upper and lower case letters in the significance markers, and according to the correct significance markers the result obtained is that the difference between each group is significant, and here I deeply apologize for my carelessness. And we have added descriptions of Nexposed and Nburied in lines 398-401, 409-411 of the manuscript. Done as requested, with changes marked in red in the manuscript. Thanks for your correction.

Point 36: Line 429 – “significantly higher than those in the control group” - And here again it is not consistent with the statistical analysis - taking into account 300W and times 0, 20 and 40 min, please analyze the description of the results again in terms of compliance with the statistical analysis.

Response 36: Thank you for reading our manuscript carefully and for your valuable advice. I apologize for my carelessness, we have double-checked and double-checked the data in Table 3 and Problems were found with the significance labeling of some of the data, allowing for errors in the description of the results. We've changed it in the manuscript and highlighted it in red. Thanks for your correction. The results obtained based on the analysis of the corrected data were that the number of β-sheets, α-helices and β-turns in the ultrasonically treated composite gel were significantly higher than in the control group.

Point 37: Table 3 - is the title good? Isn't this the title from the previous table?

Response 37: We apologize for our careless mistake and thank you for the reminder. Based on your comments, we have changed the title of table 3 to: Effect of ultrasonic power and time on the secondary structure content of bovine plasma pro-tein-carboxymethyl cellulose complex gel. Done as requested, please check line 480, 481.

Point 38: Conclusion - please modify your conclusions in accordance with the updated version of the results and discussion, which will be based on statistical analysis.

Response 38: Thank you for reading our manuscript carefully and for your valuable advice. We have modified the conclusions based on the modified results and discussion to read: In this study, the effects of ultrasonic treatment (300 W, 400 W, 500 W, 600 W, 700 W) and ultrasonic time (0 min, 10 min, 20 min, 30 min, and 40 min) on the structure and gel properties of bovine plasma protein-carboxymethylcellulose composite gels were investigated. The results showed that:ultrasonication promoted the transition of protein conformation from random curl to α-helix, β-spin and β-sheet, increased the surface hydrophobicity and free sulfhydryl content of the composite gel, and enhanced the hydrophobicity and disulfide bonding of the composite gel. The change in protein conformation and the increase in intermolecular forces led to the formation of a dense and homogeneous composite gel network, which improved the rheological properties, gel strength, and WHC of the composite gel. At the same ultrasound power, the improvement in water holding capacity, thermal stability, intermolecular forces, and secondary structure of the composite gel was most significant at 30 min. However, when the sonication time was too long (>30 min), the pores of the reticular structure of the composite gel became larger, and G′, gel strength and WHC decreased. This study showed that ultrasonication is an effective method to improve the properties of bovine plasma-carboxymethylcellulose composite gels, which provides a theoretical basis for its application in food processing. Done as requested, with changes marked in red in the manuscript.

Round 2

Reviewer 3 Report

Comments and Suggestions for Authors

Title: Effect of ultrasonic treatment on the physicochemical properties of bovine plasma protein-carboxymethyl cellulose composite gel

Liyuan Wang, Yu Ma; Ruheng Shen, Li Zhang, Long He, Yuling Qu, Xiaotong Ma, Guoyuan Ma, Zhaobin Guo, Cheng Chen, Hongbo Li and Xiangying kong

Overview and general recommendation:

In my opinion, the manuscript has been improved in accordance with my comments. There are only minor comments left for improvement, which I include below.

Minor comments:

Line 122 - Please correct the writing of the ultrasound power into capital letters “W”

Figure 3 - Below table 3, please add an explanation of what the statistical symbols mean - uppercase and lowercase letters, it is correctly coded for table 1.

Comment #33 from the previous review still needs improvement – “As shown in Figure 3. (b), when the ultrasound time increases from 0 minutes to 40 minutes, and the ultrasound power increases from 300W to 700W, the content of free sulfhydryl in the composite gel first increases and then decreases. At an ultrasound time of 30 minutes and an ultrasound power of 500W, the maximum content of free sulfhydryl in the composite gel is 9.21 mmol/g.” - this fragment is still not consistent with the results of the statistical analysis. Please remember that when values are marked with the same letters, they are the same, there is no decrease in value!

Author Response

Please find the manuscript in the attached file.

Point 1: Line 122 - Please correct the writing of the ultrasound power into capital letters “W”.

Response 1: We apologize for our careless mistake and thank you for the reminder. Based on your comments, we have changed the unit of ultrasonic power to "W" in the manuscript. Done as requested, please check line 124.

Point 2: Figure 3 - Below table 3, please add an explanation of what the statistical symbols mean - uppercase and lowercase letters, it is correctly coded for table 1.

Response 2: We apologize for our careless mistake and thank you for the reminder. Based on your comments, we have changed the interpretation of the meaning of the statistical symbols in Table 3 of the manuscript to read: lowercase letters indicate significant differences between different ultrasound powers. Upper case letters indicate significant differences between different ultrasound times. Done as requested, please check line 491 and 492.

Point 3: Comment #33 from the previous review still needs improvement – “As shown in Figure 3. (b), when the ultrasound time increases from 0 minutes to 40 minutes, and the ultrasound power increases from 300W to 700W, the content of free sulfhydryl in the composite gel first increases and then decreases. At an ultrasound time of 30 minutes and an ultrasound power of 500W, the maximum content of free sulfhydryl in the composite gel is 9.21 mmol/g.” - this fragment is still not consistent with the results of the statistical analysis. Please remember that when values are marked with the same letters, they are the same, there is no decrease in value!

Response 3: Thank you for your careful reading of our manuscript. We have changed this part of the manuscript to read that the free sulfhydryl content of the composite gels showed a gradual increase as the ultrasound time was increased from 0 to 30 min. And the free sulfhydryl content of the composite gel when the ultrasound time reached 20, 30 and 40 minutes was significantly higher than that of the composite gel without ultrasonic treatment (P < 0.05). However, the free sulfhydryl content of the composite gels with ultrasound time of 40 minutes was significantly lower (P < 0.05) than that of the samples with ultrasound time of 30 minutes when the ultrasound power was 400, 600 and 700 W. Done as requested, please check line 369-376.
